

# Combining motor imagery with action observation training does not lead to a greater autonomic nervous system response than motor imagery alone during simple and functional movements: a randomized controlled trial

Ferran Cuenca-Martínez[1,2], Luis Suso-Martí[1,2], Mónica Grande-Alonso[1,2], Alba Paris-Alemany[1,2,3,4] and Roy La Touche[1,2,3,4]

[1] Departamento de Fisioterapia, Centro Superior de Estudios Universitarios La Salle, Universidad Autónoma de Madrid, Madrid, Spain
[2] Motion in Brains Research Group, Institute of Neuroscience and Sciences of the Movement (INCIMOV), Centro Superior de Estudios Universitarios La Salle, Universidad Autónoma de Madrid, Madrid, Spain
[3] Instituto de Neurociencia y Dolor Craneofacial (INDCRAN), Madrid, Spain
[4] Instituto de Investigación Sanitaria del Hospital Universitario La Paz (IdiPAZ), Madrid, Spain

Corresponding authors
Ferran Cuenca-Martínez, fecuen2@gmail.com
Roy La Touche, roylatouche@yahoo.es

## ABSTRACT

Both motor imagery (MI) and action observation (AO) trigger the activation of the neurocognitive mechanisms that underlie the planning and execution of voluntary movements in a manner that resembles how the action is performed in a real way. The main objective of the present study was to compare the autonomic nervous system (ANS) response in an isolated MI group compared to a combined MI + AO group. The mental tasks were based on two simple movements that are recorded in the revised movement imagery questionnaire in third-person perspective. The secondary objective of the study was to test if there was any relationship between the ANS variables and the ability to generate mental motor imagery, the mental chronometry and the level of physical activity. The main outcomes that were measured were heart rate, respiratory rate and electrodermal activity. A Biopac MP150 system, a measurement device of autonomic changes, was used for the quantification and evaluation of autonomic variables. Forty five asymptomatic subjects were selected and randomized in three groups: isolated MI, MI + AO and control group (CG). In regards to the activation of the sympathetic nervous system (SNS), no differences were observed between MI and MI + AO groups ($p > .05$), although some differences were found between both groups when compared to the CG ($p < .05$). Additionally, even though no associations were reported between the ANS variables and the ability to generate mental motor imagery, moderate-strong positive associations were found in mental chronometry and the level of physical activity. Our results suggest that MI and MI + AO, lead to an activation of the SNS, although there are no significant differences between the two groups. Based on results obtained, we suggest that tasks of low complexity, providing a visual input through the AO does not facilitates their subsequent motor imagination. A higher level of physical activity as well as a longer time to perform mental task, seem to be associated with a greater increase in the ANS response.

## INTRODUCTION

Motor imagery (MI) is defined as a dynamic mental process that involves the representation of an action, in an internal way, without its actual motor execution (*Decety, 1996*). The action observation (AO) evokes an internal, real-time motor simulation of the movements that the observer perceives visually (*Rizzolatti & Sinigaglia, 2010*; *Buccino, 2014*). Both mental processes trigger the activation of the neurocognitive mechanisms that underlie the planning and execution of voluntary movements in a manner that resembles how the action is performed in a real manner (*Stephan et al., 1995*; *Luft et al., 1998*; *Lotze et al., 1999*; *Wright, Williams & Holmes, 2014*).

Both observation and imagination share a great number of common mental processes based primarily on sensory perception, and the information stored by memory systems (*Jeannerod, 2001*). The activation of the motor command during a mental practice does not induce an active movement probably due to an inhibitory mechanism in the primary motor cortex on the descending corticospinal tract pathways. Nevertheless, whereas some studies did not report primary motor cortex activations during MI, others found low involvement or relevant activation, showing some discrepancies (*Guillot & Collet, 2005a*; *Guillot et al., 2012*).

The practice of mental training involves a component of the autonomic nervous system (ANS). Autonomic activation during motor imagery appears to be centrally controlled (*Decety et al., 1993*). The sympathetic pathways to the heart are modulated by the activity of the anterior cingulate cortex, and cardiovagal activity is under the control of the ventral medial prefrontal cortex (*Wong et al., 2007*). Electrodermal activity is mediated by neural networks involving the prefrontal, insular, and parietal cortices, and limbic structures, including the cingulate and medial temporal lobe, along with the amygdala and the hippocampus (*Critchley, 2002*). The neural substrate for these peripheral autonomic responses is associated with motivational and affective states which, in turn, mediate motor imagery.

It has been shown that both MI and AO lead to changes in the ANS that cause sympathetic responses, although the neurophysiological bases remain uncertain and are still based on hypotheses (*Beyer et al., 1990*; *Decety et al., 1991*; *Lang et al., 1993*; *Thill et al., 1997*; *Bolliet, Collet & Dittmar, 2005*; *Collet et al., 2013*). The functional relations between both neurocognitive processes and the sympathetic nervous system (SNS) could be based on a preparation phase in which, the activation of the SNS, happens to a near effort and, therefore, to a close energy expenditure in physiological processes (i.e., cardiorespiratory adaptations, and anticipated adaptations in body temperature and sweat rate) which will take place in order to face said metabolic changes produced by the voluntary movement itself. In addition, several hypotheses have been described regarding the notion that the SNS not only has the quantitative objective of providing energy to the muscle effectors, but that it also qualitatively and specifically designs and adapts the parameters on demand

in an attempt to save the energy provided for each precise motor execution. (*Decety et al., 1991*; *Decety et al., 1993*; *Collet et al., 2013*).

Taking into account that both MI and AO cause ANS response that induce an increase in heart rate, blood pressure, respiratory rate and electrodermal activity (*Lang et al., 1993*; *Paccalin & Jeannerod, 2000*; *Bolliet, Collet & Dittmar, 2005*; *Papadelis et al., 2007*; *Brown, Kemp & Macefield, 2013*). Previous findings using fRMI reported that MI and AO + MI each have a unique neural signature, involving greater neural activity for AO + MI in the bilateral cerebellum compared with MI alone. In addition, in areas such as the supplementary motor cortex and the left precentral gyrus, AO + MI showed increased activity compared with MI independently (*Taube et al., 2015*). Research using multi-channel electroencephalographic (EEG) recordings has also reported more pronounced electrophysiological activity over primary sensorimotor and parietal regions in the mu/alpha and beta frequency bands for AO + MI, relative to MI in isolation (*Eaves, Haythornthwaite & Vogt, 2014*). In addition, research into observation and imagery effects using single-pulse transcranial magnetic stimulation (TMS) over the motor cortex showed that AO + MI produces significantly greater facilitation of corticospinal excitability compared with MI alone (*Sakamoto et al., 2009*). Therefore, our hypothesis is that the combination of MI and AO induces an autonomic response shift greater than MI does in isolation.

The main objective of this study was to compare the results obtained from intervention groups on the subject of the sympathetic activation of the ANS in a program that combined MI with AO, in contrast to an isolated MI program on asymptomatic subjects. The secondary objective of the present study was to explore whether there is any relationship between the ANS variables and the ability to generate motor imagery, the mental chronometry, and the level of physical activity.

## MATERIAL & METHODS

### Study design

The present study was a randomized controlled trial. The study was planned and performed in accordance with the requirements of the CONSORT (Consolidated Standards of Reporting Trials) statement (*Schulz et al., 2010*).

### Recruitment of participants

A sample of asymptomatic subjects was obtained from La Salle University and from the Community of Madrid through media and social networks, posters, brochures, and emails. The subjects were recruited between March and June of 2017. The inclusion criteria were as follows: healthy and with no pain subjects, and age between 18 and 60. The exclusion criteria included the following: (a) subjects who presented systemic, cardio-respiratory, central nervous system or rheumatic diseases, or those who presented any musculoskeletal pathology with a source of symptoms at the time of the study; (b) underage subjects; (c) subjects with pain at the time of the study; and (d) subjects who were not in full use of their mental faculties and thus were not able to perform the intervention of the study.

Informed written consent was obtained from all subjects prior to inclusion. All participants were given an explanation of the study procedures, which were planned under
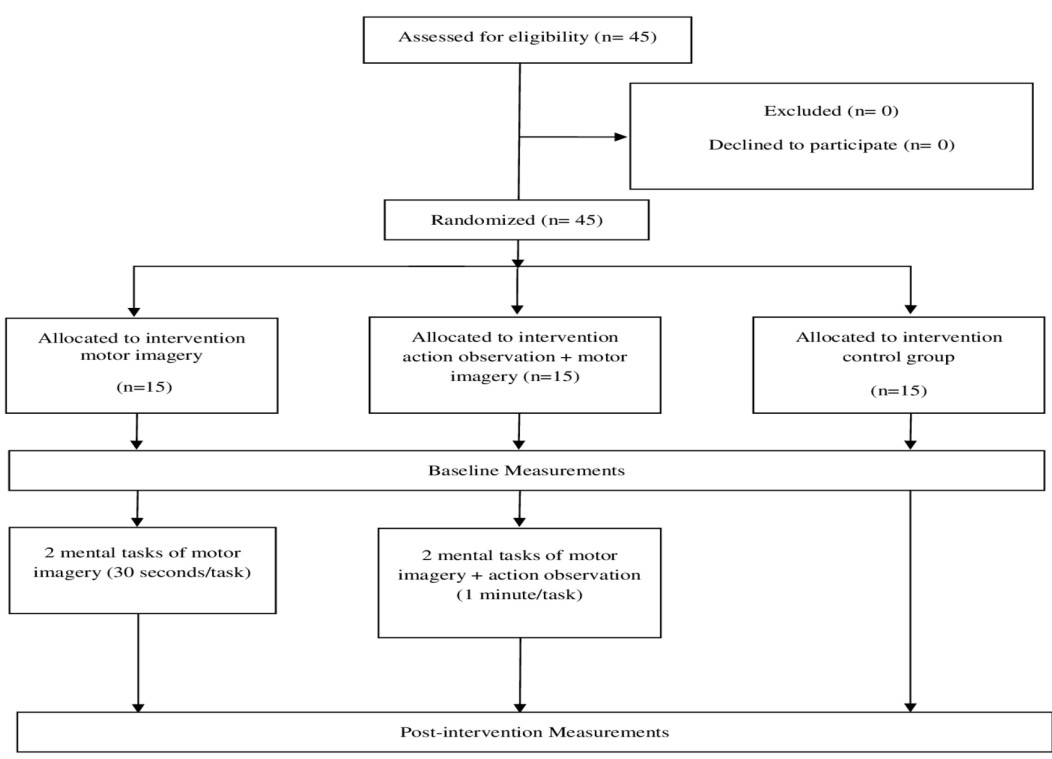

**Figure 1** **Flow chart of the study design.**

the ethical standards of the Helsinki Declaration, and which had been approved by the ethics committee of the La Salle University Center for Higher Education (CSEULS-PI-008). The trial was registered with the United States National Institutes of Health Clinical Trials Registry, with the registration number NCT03232879.

## Randomization

Randomization was performed using a computer generated random sequence table with a balanced three-block design (GraphPad Software, Inc., CA, USA). A statistician generated the randomization list, and once the initial assessment and inclusion of the participants were completed, the included participants were randomly assigned to any of the three groups using the random sequence list (MI + AO, only MI or the control group) (Fig. 1).

## Intervention
### *MI in isolation*

All the subjects in the MI group were informed of the procedure at the beginning of the intervention, which consisted of the following: first, the subjects were supine for 5 min in order to achieve a baseline condition. After that, they sat for two more minutes. Afterwards, two consecutive 30 s visual MI tasks were performed, both based on two movements that are recorded in the revised movement imagery questionnaire in third-person perspective, which is based on the representation of somebody performing a movement.
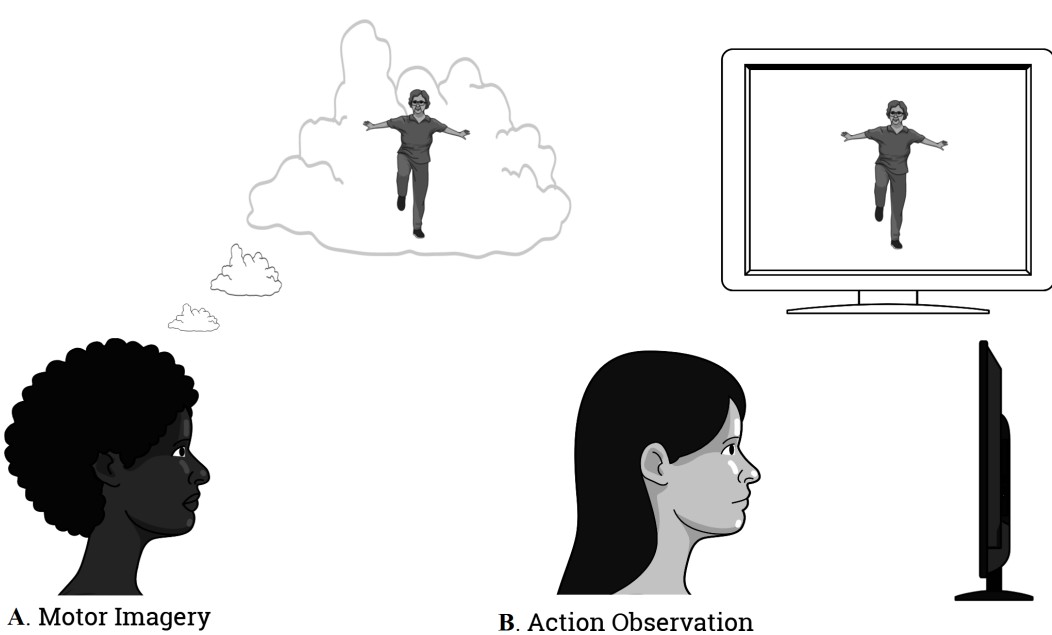

**A**. Motor Imagery

**B**. Action Observation

**Figure 2** **The illustration of the intervention.** (A) Motor Imagery; (B) Action Observation Training.

The first movement chosen for mental practice, starting from a standing up position, consisted of raising the right knee as high as possible and then returning to the starting position. The second movement, starting from a standing up position, and with arms extended along the head, consisted of bending the trunk while trying to touch their toes with their hands and then returning to the initial position. Finally, at the end of the mental training the subjects were at rest for 3 more minutes. Hence, the total time of the intervention was 11 min (Fig. 2).

### MI training combined with AO

This group, in addition to the intervention with MI previously described, underwent AO training through the following procedure: before the subjects performed the MI practice, they were presented with a 30 s video that displayed the motor task that they ought to imagine later in third-person perspective. A video was played prior to the first practice of imagination and after the second mental practice, a second video was shown. Thus, in this combined group of MI and AO the intervention lasted 12 min (Fig. 2).

### Control group

The subjects included in the control group (CG) had their neurovegetative variables recorded without having undergone any intervention, and they followed the same measurement times as the two previous groups.

### Procedures

After giving their consent to partake in the study, all the participants received a set of questionnaires prior to the intervention. These included a sociodemographic assessment, as well as an evaluation of their physical activity and of their ability to generate kinesthetic

and visual MI. The questionnaires given were the following: the Spanish validated version of the International Questionnaire of Physical Activity (IPAQ) and the Spanish validated version of the Revised Movement Imagery Questionnaire (MIQ-R). The latter was given to the participants after the intervention, where the time spent for each task of imagination was also recorded. A Biopac MP150 system, a device to measure autonomic activity, was used for the recordings and processing of the autonomic variables.

Five measurements were taken for each participant: the first of them was taken after lying down for 5 min; the second, after sitting for 2 min; the third, after finishing the first 30 s task of motor imagination; the fourth, after finishing the second 30 s task of motor imagination; and, finally, the fifth measurement was taken 3 min after finishing the mental exercise.

The Biopac MP150 system has an internal microprocessor to control the data acquisition and communication with the computer. There are 16 analog input channels, 2 analog output channels, 16 digital channels that can be used for either input or output, and an external trigger input. The digital lines can be programmed as either inputs or outputs and function in 8 channel blocks. Block 1 (I/O lines 0 through 7) can be programmed as either all inputs or all outputs, independently of block 2 (I/O lines 8 through 15).

Each condition involved a series of inter-sampling intervals (ISIs). In each condition, the first two ISIs (Pre and Pre-Pre', 420 s) was the baseline (rest) period; the next two ISIs (Pre'- $^{1st}$I and $^{1st}$I- $^{2nd}$I, 30 s each) was the intervention period. Finally, the last ISI ($^{2nd}$I-Post, 180 s) was the return to calm phase. Autonomic parameters were continuously monitored and were sampled every 30 s. In the first two ISIs, the Pre interval lasted 300 s. Ten measurements were taken, and the device calculated a mean of the 10 measurements. The Pre-Pre' interval was measured in the same way (120 s, 4 measurements). In this way, two basal measurements were obtained, the first with the participant lying down and second with the participant sitting. Then, regarding the two subsequent ISIs corresponding to the intervention period, the device averaged the values over intervals of 30 s where 5 values per second were registered during the total intervention time and an average value was automatically obtained after each 30-second interval. Thus, two measurements were taken, one every 30 s (the first relative to the first mental task and the second relative to the second). Finally, the last ISI ($^{2nd}$I-Post, 180 s) value was obtained in the same way as the baseline data. The procedure was similar to that performed by *Decety et al. (1993)*.

Previous research has shown how after MI intervention, the LF/HF ratio as an index of ANS activity returned to the resting level. Therefore, the device was programmed to record the ANS activity values concurrent with the MI session; the measurements are the mean of these values (*Bunno, Suzuki & Iwatsuki, 2015*).

All measurements were taken in a sitting position except for the first one that was taken in a lying position. The measured variables were: (1) heart rate (HR); (2) respiratory rate (RR); and (3) the electrodermal activity (EDA).

## Primary outcomes
### Autonomic nervous system variables
The main outcomes that were measured were HR, RR, and EDA. The recording of these variables was performed to objectify the changes produced in the ANS during the performance of the motor mental practice in both groups. The EDA was measured through the use of two electrodes that recorded changes in conductance through the skin located at the palmar location. They were attached with a distance of 3 cm in between thenar and hypothenar eminence of the dominant hand (*Boucsein et al., 2012*). They were disposable snap electrodes pre-gelled with isotonic gel (Ag/AgCl contact). The size was 27 mm wide ×36 mm long ×1.5 mm thick. Contact: 11 mm diameter. Gel cavity: 16 mm diameter ×1.5 mm deep. The transducer (EDA100C) operates by applying a fixed 0.5 Volts DC across the two electrodes and then measures the current flowing between the two electrodes. Given the voltage (V) is fixed, from Ohms Law, the conductance (G) will be proportional to the current (I): $G = I/V = I/0.5$ V. Circuitry then converts the detected current to a voltage so it can be measured by the MP device. The software performs the necessary scaling and unit conversion. The RR was measured through a respiratory effort transducer, SS5LB model. It was placed around the chest like a strap passing over the xiphoid process (*La Touche et al., 2013*). Finally, the HR was measured by three electrodes located in the left area of the chest: one placed one centimetre under the left sternoclavicular joint, a second one positioned two centimetres under the left acromioclavicular joint, and a third one was located five centimetres under the first electrode, in the sixth left sternocostal joint (*Niendorf, Winter & Frauenrath, 2012*). The electrodes were disposable snap electrodes pre-gelled with isotonic gel (Ag/AgCl contact). The size was 40 mm (diameter), with foam tape (1.5 mm thick) backing, and incorporated a gel cavity (16 mm diameter ×1.5 mm deep).

## Secondary outcomes
### Visual and kinesthetic motor imagery ability
MIQ-R has four movements repeated in two subscales, a visual and a kinesthetic one. Additionally, a score between 1 and 7 is assigned, with 1 representing difficulty in picturing the motor image or difficulty in feeling the movement previously made, and 7 representing the maximum ease. The internal consistencies of the MIQ-R have been consistently adequate with Cronbach's α coefficients ranging above 0.84 for the total scale, 0.80 for de visual subscale and 0.84 for the kinesthetic subscale (*Campos & González, 2010*).

## Mental chronometry
Mental chronometry evaluation was also used to measure the subject's motor imagery ability. Using a stopwatch, the time spent for performing each MIQ-R task was recorded. Time recorded corresponds to the interval between the command to start the task, given by the evaluator, and the verbal response of conclusion of the task, given by the subject. Mental chronometry is a reliable behavioral task that has previously been employed to collect an objective measure of MI ability (*Guillot & Collet, 2005b*; *Malouin et al., 2008*; *Williams et al., 2015*).

### The level of physical activity

The level of physical activity was objectified through the IPAQ questionnaire, which allows the subjects to be divided into three groups according to their level of activity, which can be high, moderate, and low or inactive (*Roman-Viñas et al., 2010*). This questionnaire has shown an acceptable validity to measure total physical activity. Therefore, the psychometric properties of the questionnaire were accepted for use in studies that required the measurement of physical activity.

### Sample size calculation

The sample size was estimated with the program G*Power 3.1.7 for Windows (G*Power© from the University of Dusseldorf, Germany) (*Faul et al., 2007*). The sample size calculation was considered as a power calculation to detect between-group differences in the primary outcome measures (skin conductance). We considered three groups and four measurements for primary outcomes to obtain 95% statistical power (1-$\beta$ error probability) with an $\alpha$ error level probability of .05 using analysis of variance (ANOVA) of repeated measures, within-between interaction, and an effect size of $\eta_p^2 = 0.25$ obtained from a pilot study conducted with eight participants (four per group). This generated a sample size of 15 participants per group (a total of 45 participants).

### Data analysis

The data analysis was performed using the Statistics Package for Social Science (SPSS 22.00, IBM Inc., Armonk, NY, USA).

For data analysis, we used a confidence interval of 95%, considering all those values that had a $p$ value of less than .05 to be statistically significant.

Descriptive statistics that were used to summarize data for continuous variables are presented as mean $\pm$ standard deviation and the 95% confidence interval, while categorical variables are presented as an absolute number or relative frequency percentage. A chi-square test with residual analysis was used to compare categorical variables. The normal distribution of all primary and secondary measures data was assessed using the Shapiro–Wilk test ($p > .05$). In order to compare the means of the ANS measures, the repeated measures ANOVA test was used. The effect size (Cohen's $d$) was calculated for the main autonomic variables. According to Cohen method, the effect was considered as small (0.20 to 0.49), medium (0.50 to 0.79), and large ($>0.80$) (*Cohen, 1988*). To conclude, the association between the autonomic response and the ability to generate motor imagery, the mental chronometry, and the level of physical activity was examined using the Pearson's correlation coefficient. A Pearson's correlation coefficient greater than 0.60 is considered to have a strong association, one between 0.30 and 0.60 indicates a moderate association, while less than 0.30 indicates a poor association (*Hinkle, Wiersma & Jurs, 1990*).

## RESULTS

A total of 45 asymptomatic subjects were included in the present study and were randomly assigned to three balanced groups consisting of 15 subjects per group. There were no adverse events or dropouts reported in any of the groups. No statistically significant differences were found in the sociodemographic data ($p > .05$) (Table 1).

**Table 1  Descriptive statistics of socio-demographic data.**

| Measures | MI group (n = 15) | MI + AO group (n = 15) | Control group (n = 15) | p value |
|---|---|---|---|---|
| Age | 37.07 ± 13.86 | 37.93 ± 13.17 | 38.47 ± 13.67 | .960 |
| Height | 166.87 ± 5.20 | 167.60 ± 6.40 | 168.27 ± 9.40 | .870 |
| Weight | 63.73 ± 15.31 | 66.80 ± 11.89 | 68.6 ± 12.45 | .768 |
| Gender | | | | .293 |
|     Male | 4 (26.7) | 5 (33.3) | 8 (53.3) | |
|     Female | 11 (73.3) | 10 (66.7) | 7 (46.7) | |
| Educational Level | | | | .399 |
|     Primary education | 1 (6.7) | 0 (0.0) | 0 (0.0) | |
|     Secondary education | 5 (33.3) | 6 (40.0) | 9 (60.0) | |
|     College education | 9 (60.0) | 9 (60.0) | 6 (40.0) | |

Notes.

Values are presented as mean ± standard deviation or number (%)

MI, Motor Imagery; MI + AO, Motor Imagery and Action Observation; CG, Control Group.

**Table 2  Descriptive statistics of self-report and baseline autonomic sympathetic-excitatory outcomes.**

| Measures | MI group (n = 15) | MI + AO group (n = 15) | Control group (n = 15) | p value |
|---|---|---|---|---|
| HR | 70.10 ± 7.79 | 66.73 ± 11.01 | 75.00 ± 11.78 | .101 |
| RR | 13.71 ± 2.34 | 13.2 ± 1.83 | 13.7 ± 1.84 | .758 |
| EDA | 0.85 ± 1.33 | 0.97 ± 1.28 | 1.34 ± 0.93 | .510 |
| MIQ-R | | | | |
|     MIQR-K | 24.27 ± 3.34 | 24.47 ± 3.29 | 24.27 ± 3.82 | .984 |
|     MIQR-KT | 13.91 ± 9.39 | 15.49 ± 9.10 | 9.38 ± 4.75 | .109 |
|     MIQR-V | 25.00 ± 3.04 | 25.20 ± 4.03 | 23.40 ± 4.43 | .388 |
|     MIQR-VT | 12.26 ± 8.18 | 13.62 ± 7.68 | 8.99 ± 4.20 | .182 |
| IPAQ | | | | .877 |
|     Low level | 2 (13.3) | 1 (6.7) | 3 (20.0) | |
|     Moderate level | 7 (46.7) | 8 (53.3) | 7 (46.7) | |
|     High level | 6 (40.0) | 6 (40.0) | 5 (33.3) | |

Notes.

Values are presented as mean ± standard deviation or number (%)

MI, Motor Imagery; MI + AO, Motor Imagery and Action Observation; CG, Control Group; HR, Heart Rate; RR, Respiration Rate; EDA, Electrodermal Activity; MIQ-R, the Revised Movement Imagery Questionnaire; MIQR-K, Kinesthetic subscale; MIQR-KT, Time employed in Kinesthetic subscale; MIQR-V, Visual subscale; MIQR-VT, Time employed in Visual subscale; IPAQ, International Physical Activity Questionnaire.

No statistically significant differences were found in any of the baseline measurements between three groups ($p > .05$) (Table 2). All primary and secondary measures presented a normal distribution ($p > .05$).

## Primary outcomes
### Heart rate

The ANOVA revealed significant changes in HR during group*time ($F = 16.81$, $p < .01$, $\eta^2 = .445$) and time ($F = 45.61$, $p < .01$, $\eta^2 = .521$). The post hoc analysis revealed significant inter-group differences between the two intervention groups compared to the CG with a large effect size ($p < .01$, $d > 0.8$), although there were no significant differences

between them ($p > .05$). The analysis also revealed statistically significant intra-group differences in both the MI group and the MI + AO group in every comparison with a large effect size ($p < .05$, $d > 0.8$), which did not occur in the CG ($p > .05$) (Table 3).

*Respiration rate*
The ANOVA revealed a significant interaction in group*time ($F = 29.82$, $p < .01$, $\eta^2 = .587$), and in time ($F = 85.57$, $p < .01$, $\eta^2 = .671$). The post hoc analysis revealed significant inter-group differences between the two intervention groups compared to the CG with a large effect size ($p < .01$, $d > 0.8$), although there were no significant differences between the two groups ($p > .05$). The analysis also revealed statistically significant intra-group differences in both the MI group and the MI + AO group in every comparison with a large effect size ($p < .05$, $d > 0.8$), which was slightly higher in the combined group, and there were no significant differences in the CG ($p > .05$) (Table 3).

*Electrodermal activity*
There were significant differences in time ($F = 3.95$, $p < .05$, $\eta^2 = .08$), and in group* time ($F = 6.65$, $p < .01$, $\eta^2 = .241$). No statistically significant differences were found between the two intervention groups ($p > .05$), but some differences were encountered between both groups and the CG, in the first and second interventions, compared to the initial measure with a large effect size for both groups ($p < .05$, $d > 0.8$), although this was not the case in the return to calm ($p > .05$). At the intra-group level, both the combined MI + OA and MI alone showed significant changes between the two interventions and the initial measurement, while these changes were not found in the CG ($p > .05$) (Table 4). All values are in Table 5.

## Secondary outcomes
In the correlation analysis, no association was found between the ANS response and the ability to generate motor images ($p > .05$). However, regarding mental chronometry, it was found that there is a moderate-strong positive association in the MI group with respect to HR increase ($p < .05$, $r = .643$), RR and the time spent for performing each MIQ-R task ($p < .05$, $r = .575$). Regarding the level of physical activity, based on the data obtained, there appears to be a moderate-strong association between a higher IPAQ score and a higher HR in the MI ($p < .05$, $r = .559$) and MI + AO ($p < .05$, $r = .621$) groups, even though no correlation was found in the other variables.

## DISCUSSION

The main objective of this study was to observe whether an MI intervention, together with in AO training, caused an increase in the ANS response greater than a MI intervention in isolation in asymptomatic subjects. The findings suggest that in both intervention groups (IM and IM + OA) there is an increase in the vegetative response compared to the CG but there were no significant differences between the two groups in all ANS measures.

The findings obtained in the present study show controversy regarding the current state of the art, in which a large number of studies have shown that the combined practice of MI, together with AO, cause an increase in neurophysiological activity greater than the isolated

**Table 3  Comparative analysis of the primary outcomes.**

| Measure | Group | Mean ± SD | | | | Mean difference (95% CI); effect size ($d$) |
|---|---|---|---|---|---|---|
| | | **Pre'-Pre** | **1st I-Pre** | **2nd I-Pre** | **Post-2nd I** | **(a)** (Pre'-Pre) vs. (1st I-Pre) **(b)** (Pre'-Pre) vs. (2nd I-Pre) **(c)** (2nd I-Pre) vs. (Post-2nd I) |
| Heart rate (Hpm) | MI | $-0.88 \pm 4.70$ | $6.15 \pm 3.43$ | $9.47 \pm 5.32$ | $-6.29 \pm 5.35$ | (a) $-7.04^*$ ($-11.62$ to $-2.46$); $d=-1.71$ (b) $-10.35^{**}$ ($-15.23$ to $-5.47$); $d=-2.06$ (c) $5.40^*$ ($1.34$ to $9.46$); $d=2.95$ |
| | MI + AO | $-0.23 \pm 2.68$ | $6.80 \pm 6.32$ | $10.10 \pm 5.88$ | $-7.26 \pm 5.28$ | (a) $-7.04^*$ ($-11.61$ to $-2.47$); $d=-1.44$ (b) $-10.34^{**}$ ($-15.22$ to $-5.46$); $d=-2.26$ (c) $7.03^{**}$ ($2.97$ to $11.09$); $d=3.10$ |
| | CG | $1.46 \pm 3.04$ | $-1.40 \pm 4.03$ | $-2.20 \pm 3.56$ | $-0.66 \pm 2.12$ | (a) $2.86$ ($-1.71$ to $7.43$); $d=0.80$ (b) $3.66$ ($-1.21$ to $8.54$); $d=1.10$ (c) $2.21$ ($-1.92$ to $6.19$); $d=-0.52$ |
| Mean difference (95% CI); effect size ($d$) | MI vs. MI + AO | $-0.65$ ($-3.91$ to $2.61$); $d=-0.17$ | $-0.65$ ($-4.98$ to $3.68$); $d=-0.12$ | $-0.63$ ($-5.21$ to $3.93$); $d=-0.11$ | $0.97$ ($-3.13$ to $5.08$); $d=0.18$ | |
| | MI vs. CG | $-2.35$ ($-5.62$ to $0.91$); $d=-0.59$ | $7.55^{**}$ ($3.21$ to $11.89$); $d=2.01$ | $11.67^{**}$ ($7.09$ to $16.24$); $d=2.57$ | $-5.62^*$ ($-9.73$ to $-1.51$); $d=-1.38$ | |
| | MI + AO vs. CG | $-1.70$ ($-4.96$ to $1.56$); $d=-0.58$ | $8.20^{**}$ ($3.86$ to $12.54$); $d=1.54$ | $12.30^{**}$ ($7.73$ to $16.88$); $d=2.53$ | $-6.60^*$ ($-10.71$ to $-2.49$); $d=-1.64$ | |
| | | **Pre'-Pre** | **1st I-Pre** | **2nd I-Pre** | **Post-2nd I** | **(a)** (Pre'-Pre) vs. (1st I-Pre) **(b)** (Pre'-Pre) vs. (2nd I-Pre) **(c)** (2nd I-Pre) vs. (Post-2nd I) |
| Respiration Rate (Bpm) | MI | $0.06 \pm 1.01$ | $2.67 \pm 1.97$ | $4.34 \pm 2.17$ | $-4.19 \pm 2.08$ | (a) $-2.61^*$ ($-4.47$ to $-0.74$); $d=-1.66$ (b) $-4.28^{**}$ ($-6.19$ to $-2.37$); $d=-2.52$ (c) $8.54^{**}$ ($6.10$ to $10.98$); $d=4.01$ |
| | MI + AO | $0.22 \pm 0.73$ | $3.19 \pm 2.21$ | $5.63 \pm 2.03$ | $-4.53 \pm 2.14$ | (a) $-2.96^{**}$ ($-4.83$ to $-1.09$); $d=-1.80$ (b) $-5.40^{**}$ ($-7.31$ to $-3.49$) $d=-3.54$ (c) $10.16^{**}$ ($7.72$ to $12.60$); $d=4.87$ |
| | CG | $0.12 \pm 1.83$ | $-0.46 \pm 1.50$ | $-0.82 \pm 1.79$ | $0.17 \pm 1.26$ | (a) $0.58$ ($-1.28$ to $2.44$); $d=0.34$ (b) $0.94$ ($-0.97$ to $2.85$); $d=0.51$ (c) $-0.99$ ($-3.43$ to $1.44$); $d=-0.63$ |

Peer J

**Table 3** (*continued*)

| Measure | Group | Mean ± SD | | | Mean difference (95% CI); effect size (*d*) |
|---|---|---|---|---|---|
| | MI vs. MI + AO | −0.16 (−1.33 to 1.00); *d* = −0.18 | −0.52 (−2.27 to 1.23); *d* = −0.24 | −1.28 (−3.11 to 0.54); *d* = −0.61 | 0.33 (−1.37 to 2.03); *d* = 0.16 |
| Mean difference (95% CI); effect size (*d*) | MI vs. CG | −0.06 (−1.22 to 1.10); *d* = −0.04 | 3.13** (1.38 to 4.88); *d* = 1.78 | 5.16** (3.33 to 6.99); *d* = 2.59 | −4.37** (−6.08 to −2.66); *d* = −2.53 |
| | MI + AO vs. CG | 0.10 (−1.06 to 1.27); *d* = 0.07 | 3.65** (1.90 to 5.40); *d* = 1.93 | 6.45** (4.62 to 8.27); *d* = 3.37 | −4.70** (−6.41 to −2.99); *d* = −2.62 |

**Notes.**

*$p < .05$

**$p < .01$

SD, Standard Deviation; vs., versus; (Pre'-Pre), Pre-Intervention Differences; ([1st]I-Pre), Differences between First Intervention and Pre-Intervention; ([2nd]I-Pre), Differences between Second Intervention and Pre-Intervention; (Post- [2nd]I), Differences between Post-Intervention and Second Intervention; MI, Motor Imagery; MI + AO, Motor Imagery and Action Observation; CG, Control Group; Hpm, heartbeats per minute; Bpm, breaths per minute.

Cuenca-Martínez et al. (2018), *PeerJ*, DOI 10.7717/peerj.5142

**Table 4  Comparative analysis of the primary outcomes.**

| Measure | Group | Mean ± SD | | | | Mean difference (95% CI); effect size (d) |
|---|---|---|---|---|---|---|
| | | Pre'-Pre | 1st I-Pre | 2nd I-Pre | Post-2nd I | (a) (Pre'-Pre) vs. (1st I-Pre) (b) (Pre'-Pre) vs. (2nd I-Pre) (c) (2nd I-Pre) vs. (Post-2nd I) |
| Electrodermal activity (μS) | MI | −0.09 ± 0.22 | 0.13 ± 0.31 | 0.24 ± 0.56 | 0.10 ± 0.21 | (a) −0.23* (−0.46 to −0.02); $d = -0.81$ (b) −0.33* (−0.63 to −0.03); $d = -0.77$ (c) 0.13 (−0.07 to 0.34); $d = 0.33$ |
| | MI + AO | −0.06 ± 0.12 | 0.17 ± 0.26 | 0.23 ± 0.34 | 0.08 ± 0.11 | (a) −0.24* (−0.47 to −0.19); $d = -1.13$ (b) −0.30* (−0.60 to −0.04); $d = -1.13$ (c) 0.15 (−0.05 to 0.36); $d = 0.59$ |
| | CG | 0.06 ± 0.16 | −0.12 ± 0.19 | −0.15 ± 0.19 | 0.00 ± 0.10 | (a) 0.18 (−0.04 to 0.40); $d = 1.02$ (b) 0.21 (−0.09 to 0.51); $d = 1.19$ (c) −0.15 (−0.36 to 0.05); $d = -0.98$ |
| Mean difference (95% CI); effect size (d) | MI vs. MI + AO | −0.02 (−0.18 to 0.13); $d = -0.16$ | −0.04 (−0.28 to 0.20); $d = -0.14$ | 0.00 (−0.36 to 0.37); $d = 0.02$ | 0.02 (−0.11 to 0.16); $d = 0.11$ | |
| | MI vs. CG | −0.15 (−0.3 to 0.07); $d = -0.78$ | 0.25* (0.01 to 0.49); $d = 0.97$ | 0.39* (0.03 to 0.76); $d = 0.93$ | −0.10 (−0.03 to 0.24); $d = 0.60$ | |
| | MI + AO vs. CG | −0.13 (−0.29 to 0.32); $d = -0.84$ | 0.29* (0.06 to 0.54); $d = 1.27$ | 0.39* (0.02 to 0.75); $d = 1.38$ | 0.08 (−0.05 to 0.22); $d = 0.76$ | |

**Notes.**

*$p < .05$.

**$p < .01$.

SD, Standard Deviation; vs., versus; (Pre'-Pre), Pre-Intervention Differences; (1st I-Pre), Differences between First Intervention and Pre-Intervention; (2nd I-Pre), Differences between Second Intervention and Pre-Intervention; (Post- 2nd I), Differences between Post-Intervention and Second Intervention; MI, Motor Imagery; MI + AO, Motor Imagery and Action Observation; CG, Control Group; μS, Microsiemens.

**Table 5  Mean values of the dependent variables.**

| Measure | Group | Mean ± SD | | | | |
|---|---|---|---|---|---|---|
| | | Pre | Pre' | 1st I | 2nd I | Post |
| | MI | 70.10 ± 7.79 | 69.51 ± 7.90 | 75.67 ± 6.69 | 78.98 ± 7.75 | 72.69 ± 7.38 |
| Heart rate (Hpm) | MI + AO | 66.73 ± 11.01 | 66.19 ± 11.89 | 73.00 ± 11.83 | 76.30 ± 11.64 | 69.03 ± 14.27 |
| | CG | 75.00 ± 11.04 | 76.46 ± 12.68 | 75.06 ± 11.90 | 74.26 ± 11.24 | 73.60 ± 10.69 |
| | | Pre | Pre' | 1st I | 2nd I | Post |
| | MI | 13.71 ± 2.34 | 13.95 ± 2.10 | 16.62 ± 2.23 | 18.29 ± 1.87 | 14.09 ± 2.16 |
| Respiration rate (Bpm) | MI + AO | 13.20 ± 1.83 | 13.31 ± 1.66 | 16.50 ± 2.16 | 18.94 ± 1.16 | 14.41 ± 2.44 |
| | CG | 13.70 ± 1.84 | 13.88 ± 2.05 | 13.42 ± 1.86 | 13.06 ± 1.62 | 13.23 ± 1.64 |
| | | Pre | Pre' | 1st I | 2nd I | Post |
| | MI | 0.85 ± 1.33 | 0.80 ± 1.23 | 0.94 ± 1.51 | 1.04 ± 1.74 | 1.15 ± 1.94 |
| Electrodermal activity (μS) | MI + AO | 0.97 ± 1.28 | 0.86 ± 1.08 | 1.04 ± 1.31 | 1.09 ± 1.38 | 1.18 ± 1.43 |
| | CG | 1.34 ± 0.93 | 1.40 ± 0.90 | 1.28 ± 0.92 | 1.25 ± 0.93 | 1.25 ± 0.95 |

**Notes.**

SD, Standard Deviation; Pre and Pre', Pre-Intervention; 1st I, First Intervention; 2nd I, Second Intervention; Post, Post-Intervention; MI, Motor Imagery; MI + AO, Motor Imagery and Action Observation; CG, Control Group; Hpm, heartbeats per minute; Bpm, breaths per minute; μS, microsiemens.

MI (*Sakamoto et al., 2009*; *Vogt et al., 2013*). A possible answer to this finding could be given in relation to the movements chosen for mental motor practice. Both movements have three principles in common: they are habitual motor gestures with a low complexity, they do not require a great mental effort in order to carry out the training, and they are not tasks of high intensity.

## Complexity of movement

The movements chosen to carry out the mental practice in the present study were movements that were easy to perform, comfortable, daily, and at a neurophysiological level, they were movements with a broad subcortical participation, minimizing the requirements of cortical activity, which included raising the right knee as high as possible and bending the trunk while trying to touch the toes with the hands. The data revealed that all the groups started from medium-high levels of physical activity. Several studies have suggested that higher levels of physical activity are associated with a greater ability to generate mental motor images (*Robin et al., 2007*; *Di Corrado, Guarnera & Quartiroli, 2014*). The hypotheses to be able to explain this finding reside in relation to the differences in perception and somatosensory afferent integration of the body as well as the processing and control of the voluntary movement, neurophysiological mechanisms widely related to the ability to generate mental motor images. Therefore, the authors of the present study hypothesize that the movements chosen for mental practice did not present any complexity for the participants and they were widely automated. In relation to this, *Kuhtz-Buschbeck et al. (2003)* used functional magnetic resonance imaging (fMRI) to observe cortical activity during MI while performing simple and high-complexity tasks. They found that at the neurophysiological level, premotor, posterior parietal and cerebellar regions were significantly more active during motor imagery of complex movements than during mental rehearsal of the simple task.

*Demougeot et al. (2009)* obtained that imagined wrist cyclic movements, which are easy to visualise and have a very low intensity, did not provoke any variation in the HR compared to the control group, however, during mental simulation of greater complexity, such as trunk or leg movements against gravity, physiological parameters (e.g., AP and HR) significantly increased. In this study, digital motor gestures are performed even less intensely than those used in our study, which may be the explanation of the results obtained. Moreover, it has been demonstrated that MI combined with AO training is able to achieve a motor learning greater than the two interventions in an isolated way (*Smith & Holmes, 2004*; *Wright & Smith, 2009*; *Eaves et al., 2016*). *Holmes & Calmels (2008)* argued that in tasks of high complexity, providing a visual input through the AO facilitates their subsequent motor imagination, as it eliminates the need for visual imagination, allowing, therefore, the attentional focus of the imagery to concentrate on the kinesthetic subdomain.

A possible answer to this finding could be that imagined movements, the visual input that is obtained after AO training in a widely integrated and automated motor gesture, does not provide relevant visual information that facilitates the later imagination of the gesture, and therefore both the combined group and the group with the MI training in isolation undergo the same changes concerning the vegetative response. Thus, in more complex motor tasks, AO training in combination with MI would cause a greater autonomic response than MI in isolation. Therefore, movement complexity is a limited factor in the present study; more complex movements should have been used in the present study to verify this finding.

## Mental effort and intensity

*Decety et al. (1991)* found that the self-representation of walking activated the SNS, and that both HR and RR varied with the degree of mental effort of the representation as well as with the intensity of the action during the mental simulation of locomotion. In addition, *Paccalin & Jeannerod (2000)* obtained that the RR in subjects that were visualizing people running on a treadmill increased as the intensity of the observed exertion did. Therefore, the intensity of the changes caused in the ANS is closely related to the observed effort, which is greater when the action requires a greater effort.

*Bolliet, Collet & Dittmar (2005)* found that participants, when they observed squats at both 50% and 90% of the personal best mark, had ANS responses that varied as a function of movement intensity: autonomic responses recorded during movement observation at 90% were significantly greater and longer than those recorded during movement observation at 50%. They concluded that autonomic responses were strongly linked to the amount of observed effort. These data could be correlated with those obtained in the present study, although it would have been necessary to include exercises that require greater effort to obtain stronger conclusions. Finally, regarding mental effort, a recent study performed by *Wriessnegger et al. (2017)* examined, at a cortical level, force-related hemodynamic changes during the performance of a motor execution and MI task by means of multichannel functional near-infrared spectroscopy. It was found that the most difficult mental task, which needed greater mental effort to perform, led to higher oxygen-hemoglobin concentration changes.

## Electrodermal activity: unique innervation

Electrodermal activity is one of the most studied and relevant measures when assessing the activation of the SNS. The reason for this assertion is that the variations in EDA arise from the activity of the sweat glands, which have an exclusively sympathetic innervation. This is an exception, since usually the sympathetic responses can be modulated by a parasympathetic response, as it occurs with the HR or the RR. However, as for the EDA, this double innervation does not occur, and the decrease in the activity of the sweat glands is produced by the cessation of sympathetic innervation (*Shields et al., 1987*). Consequently, it is suspected that the increase in the EDA supposes a physiological change in SNS prior to the accomplishment of a movement, and that increase might be preceded by a cortical level processing (*Vissing, Scherrer & Victor, 1991*; *Vissing & Hjortsø, 1996*; *Critchley, 2002*). The results here in obtained for the EDA follow in agreement with the findings described in the scientific literature since *Di Rienzo et al. (2015)* and *Oishi, Kasai & Maeshima (2000)* reported increases in the electrodermal response after the completion of a protocol of motor imagery.

It should be noted, however, that no significant differences were found between the two intervention groups in the results of the EDA. This may be due to the fact that, as *Tremayne & Barry (2001)* found, the increases in EDA depend on the cortical regulation of the ANS and, hence, on the planning of the energy expenditure to be performed depending on the needs of the task being executed. The reason for this difference in the values obtained could be due to the fact that the movements included in the protocol of our study are of low intensity and widely integrated in subcortical structures, and said tasks would not require any energy expenditure or any great previous electrodermic preparation.

In addition, *Guillot et al. (2008)* based on simple movements, concluded that subjects who had not been trained in a motor imagery process had lower values in the EDA compared to subjects who had been trained prior to the imaging process. This result suggests the importance not only of the type of task, but also of the mental effort and the ability to generate motor imagery of the subjects.

## Ability to imagine and level of physical activity

The secondary objective of this research was to observe if there were any relationships between the autonomic response and the ability to generate MI, the mental chronometry, and the level of physical activity.

The scores of both the MIQ-R and the mental chronometry were used with the aim of assessing the ability to generate both kinesthetic and visual MI of the subjects of study. The scores found on the MIQ-R questionnaire were high in all groups, and there were no differences between them, hence showing that the subjects had a great ability to generate MI. The results obtained in the study conducted by *Peixoto Pinto et al. (2017)* and *Guillot et al. (2008)* were similar to those obtained in the present study. Nonetheless, no correlation was found between the autonomic response and the ability to generate motor imagery. Concerning mental chronometry, the study of correlations showed that the longer the time spent in the practice, the greater the autonomic response obtained. Thus, we can agree that mental effort and attention seem to be very closely related to autonomic response.

Regarding the level of physical activity, the data revealed that all the groups started from medium-high levels of physical activity, with no differences between them. A high level of physical activity correlates with a higher ANS response, as reported by *Oishi, Kasai & Maeshima (2000)*. Their study compared professional and novice skaters and it was shown that there was a higher increase in HR and RR in higher levels of physical activity. In addition, *Di Corrado, Guarnera & Quartiroli (2014)* found that athletes had higher mean scores on imagery ability than the non-athlete group, suggesting the relationship between physical activity and the ability to generate motor imagery. A moderate-strong association was obtained in both intervention groups, between the HR and the level of physical activity, but neither in the RR nor in EDA, probably because of the characteristics of the motor gestures used for the mental motor task.

## Clinical implications

The results of this research must be interpreted carefully because the study was conducted with healthy participants. It is not possible to completely extrapolate the results to patients who have pain or functional disorders. Despite this, the movements used in the present study, lifting the leg and bending the trunk, are widely used in a clinical setting. Thus, if the clinician wants to perform a mental practice of a simple movement, it is not necessary to perform AO training in addition to MI training, thus optimizing the treatment time.

Currently, it is unknown whether MI and AO might play a role as a cognitive tool in rehabilitation. It is therefore relevant to the theoretical explanation of MI and AO to increase knowledge about ANS activity. These findings could be useful in the therapeutic use of MI and AO as cognitive tools in rehabilitation, providing further clues about movement complexity and intensity for clinical use.

ANS activation during mental practice could provide useful indications to clinicians for the level of effort that patients develop during rehabilitation by means of mental training. This can help to optimize mental training and to minimize discomfort or fatigue for patients, as *Demougeot et al. (2009)*, argued.

MI therefore enables the practice of movements without the need to physically perform them, and has been widely used in the training of technical skills to athletes and musicians, as well as in neurorehabilitation (*Calmels et al., 2006*). In the field of sports fitness, numerous research studies have suggested that MI accelerates and improves sports performance and the learning of motor skills. Mental practice has also been shown to improve (at the psychosocial level) athletes' motivation and confidence, thereby reducing the anxiety inherent in a competitive event (*Guillot & Collet, 2008*; *Ridderinkhof & Brass, 2015*). It could be used clinically to improve the psychosocial sphere of patients through a mental practice adapted to them.

Finally, MI could be considered a complement to physical training because the combination of the two has been widely shown to be more effective in terms of performance than isolated physical training (*Feltz & Landers, 1983*). Thus, MI could have an impact on rehabilitation to improve performance in patients seeking to achieve physical goals.

## Limitations

One limitation of our study was that a movement of greater complexity and intensity could have been included to be able to compare the results between groups and to verify their influence in relation to the response of the ANS. Movement complexity should be considered as a dependent variable to be tested in future research. We also consider that it is necessary to evaluate the ANS activity evoked by mental practice according to the different levels of previous physical activity.

Moreover, our results do not allow us to isolate the effects of only AO in ANS activity. It could be interesting in future research to perform an isolated AO intervention to test how it influences ANS activity. Another limitation was the impossibility of performing masking in the present study. Finally, a placebo intervention was not performed for the CG. It would have been interesting to record the changes in a placebo intervention for the CG.

## CONCLUSIONS

Our results suggest that MI, both in isolation and in combination with AO training, trigger the excitatory activation of the SNS, although there are no significant differences between the two intervention groups when using two simple and daily movements that are recorded in the revised movement imagery questionnaire in third-person perspective.

Based on the results obtained, a higher level of physical activity seems to be associated to a higher increase in heart rate. The same occurs with mental chronometry, which seems to show a greater sympathetic activation the longer the time invested in the task and, thus, the greater the mental effort invested on it. Lastly, there seems to be no association between the ability to generate motor imagery and the response of the ANS based on the revised movement imagery questionnaire.

Future research is necessary in order to be able to contrast these results and to allow a better understanding of how the ANS operates and its relation to mental practice.

### Funding

The authors received no funding for this work.

### Competing Interests

The authors declare there are no competing interests.

### Author Contributions

- Ferran Cuenca-Martínez conceived and designed the experiments, performed the experiments, analyzed the data, contributed reagents/materials/analysis tools, prepared figures and/or tables, authored or reviewed drafts of the paper, approved the final draft.
- Luis Suso-Martí and Roy La Touche conceived and designed the experiments, analyzed the data, contributed reagents/materials/analysis tools, prepared figures and/or tables, authored or reviewed drafts of the paper, approved the final draft.

- Mónica Grande-Alonso and Alba Paris-Alemany conceived and designed the experiments, analyzed the data, contributed reagents/materials/analysis tools, authored or reviewed drafts of the paper, approved the final draft.

## Human Ethics

The following information was supplied relating to ethical approvals (i.e., approving body and any reference numbers):

All experimental procedures were approved by the La Salle University Ethics Committee in accordance with the Helsinki Declaration (CSEULS-PI 008).

## Clinical Trial Ethics

The following information was supplied relating to ethical approvals (i.e., approving body and any reference numbers):

The experimental procedure was ratified by the La Salle University Ethics Committee in accordance with the Helsinki Declaration.

## Data Availability

The raw data are provided in a Supplemental File.

## Clinical Trial Registration

The following information was supplied regarding Clinical Trial registration:

ClinicalTrials.gov: NCT03232879.

## Supplemental Information

Supplemental information for this article can be found online at http://dx.doi.org/10.7717/peerj.5142#supplemental-information.

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
