# Peer review of "Combining motor imagery with action observation training does not lead to a greater autonomic nervous system response than motor imagery alone during simple and functional movements: a randomized controlled trial"

_PeerJ, doi:10.7717/peerj.5142_

## Round 0.1 · original submission · Major Revisions

Dear Authors, There are many major issues including methodology of the experiment as highlighted by the peer reviewers.We hope that a major overhaul can be done as the same peer reviewers have explicitly informed that if these issues are not resolved the manuscript may not received positive feedback in the next round.

·

Basic reporting

No comment

Experimental design

The main concern is related to methodological concerns.
Details about autonomic nervous system (ANS) recordings should be provided.
More, the way in which these dependent variables are computed should be given. This is a strong lack in the method section.

Validity of the findings

This is dependent upon complementary information about methods of ANS data recordings and processing.

Additional comments

The aim of the study was to compare motor imagery (MI) efficiency to the association of MI and action observation (AO). The dependent variables were mainly related to autonomic nervous system recordings, i.e. heart rate, respiratory rate and electrodermal activity.
Results did not provide any evidence of a gain due to the association of MI and AO. The authors interpreted their results as a consequence of movement difficulty. As movement was very easy to perform, AO did not help the construct of the mental image of the movement and thus, did not change autonomic activity.

General comments:
Line 70: … due to an inhibitory mechanism in the primary motor cortex on the descending corticospinal tract pathways. However, this inhibition is not complete, for it is well known that the training of mental practice involves a component of the autonomic nervous system (ANS) (Guillot and Collet, 2005a).
Do you have any reference which could show where does the ANS activity originate? You mention that inhibitory mechanisms come from the primary motor cortex and it seems implicit that this only refers to somatic commands. TO prevent any misunderstanding, I think that you should give additional information about the origins of ANS commands and why this is not inhibited. To sum, inhibitory processes seems different according to the origin of the command, i.e. from somatic or vegetative centres.

Line 75 and after (80, 84, 93, 95 and 154): … changes in the ANS that cause excitatory sympathetic responses,
I would suggest to remove ‘excitatory’ when referring to the sympathetic system as it is well known that it mobilizes the resources of the organism to face energy demands, whereas the parasympathetic branch is a moderator system. You should also refer to sympathetic nervous system with caution when you describe ANS commands as these also include those from the parasympathetic branch (e.g. those controlling the respiratory system and the cardiac system as well). I suggest to use a more general terminology when you describe the commands originating from the ANS, which are not restricted to the sympathetic branch, e.g. neurovegetative variables or more generally physiological variables as your experiment only involve physiological recordings.

Line 165-167: A Biopac MP150 system, a measurement device of autonomic changes, was used for the quantification and evaluation of autonomic variables.
I would suggest slightly changes in this sentence. A Biopac MP150 system, a measurement device of autonomic activity, was used for the recordings and processing of autonomic variables.

Page 7: Autonomic system variables. I would suggest to give additional information about sensors used for the recording of EDA and HR. It is important for the reader to know that the recordings were made with respect to the international guidelines. Please, give information about EDA sensors diameter, surface, diameter. It is necessary to give the recording method, constant current or constant voltage with their features to ensure that current density did not exceed the international standard. Give also information about the method for HR recordings.
L250 -252: It is not really surprising that you evidenced no significant difference among the EDA baseline level of the 3 groups. EDA is very specific to each individual and can vary from 1.5 µS to 70 µS (or from 15 to 700 k). Thus, the dispersion within a given distribution is very high and has little chance to be statistically different from another distribution. You may try to normalize your data and make the comparison again to make you data processing more reliable. However, I acknowledge that there no actual reason to find a difference among your three experimental groups.

Line 255: changes in HR. I suppose that you refer to mean heart rate. However, the way in which you extract your values from the initial signal has not been described. I suggest that you could emphasize the method section by providing the exact dependent variables you extracted from each physiological signal and how you processed each variable.
This may also apply these principles to respiratory rate and EDA. It is not clear to me whether your comparisons were made from mean values calculated within a determined period of time (or from any other method you should specify). You should give this information to make the reader aware of you own methods of data processing.
In the discussion section (Lines 385-388) you refer to the studies by Peixoto et al. (2017) and Guillot et al. (2008). You reported that the variables they used were similar to those of your studies. However, they processed autonomic responses whereas you did not give detailed information about the way you processed your data. I supposed that you only processed mean values computed on a given period of time (i.e. the duration necessary to elaborate the mental image of the knee movement) but this was not clear to me. If now you refer to autonomic responses, this refers to a different data processing of ANS variables. In this context, what about the HR responses and the RR responses?
L 284: no association was found between the ANS response and the ability to generate motor images (p > .05). It seems that this finding contradict several previous publication showing that ANS activity during MI was different between 2 groups of participants made of individuals with high vs low MI abilities. I suggest that this difference comes from methodological concerns and I advise the authors again to give more information about the way in which they processed their data.

Discussion section:
L 296: I suggest that you may avoid using sympathetic excitatory response when you refer to general ANS variations. As previously underlined, you should keep “sympathetic response” to the variables which are only on the control of the sympathetic branch i.e. EDA.
L 300 – 316: You provided evidence that action observation associated with motor imagery did not elicit stronger ANS responses by comparison with MI alone. You considered the following hypothesis: the movement that were asked to observe are very simple (extracted from movements used in the MIQ-R test). Thus, action observation does not bring additional information in the understanding of what should be actually mentally represented and does not really help MI. This could be mention as one of the strong argument of the discussion section. As a consequence, you should question the role of AO + MI in more complex motor skill and underline movement complexity as a limited factor of your experiment. I mean that complementary data should consider movement complexity as a dependent variable to be tested.
The main limitation of this study comes from a lack of information about the methods of data processing. You should really improve this section by giving information related to the variables you really processes unless every comparison of your data with those from previous studies could be discussed and contradicted.
The other concern is related to the new knowledge brought by your study. You mentioned that the association of AO with MI did not bring additional effect (by comparison to MI alone). You interpreted this result by referring to movement complexity. I wonder whether this finding is enough innovative to be worthy to be published. As previously suggested, I wonder whether this experiment would bring additional scientific value by including an independent variable related to movement complexity.

·

Basic reporting

Describe the introduction more detail why did you make a hypothesis in this study. MI and AO can increase the ANS activity. However, in introduction, there is no previous literature about the difference between MI+AO and only MI trial. Therefore, I suggest that you provide more neurophysiological information (i.e. PET, fMRI, NIRS, and TMS) during MI+AO and MI trial.

Experimental design

In present study, there are no significant difference in various ANS indices during MI+AO and MI only trial. However, in this study, you did not study about the ANS activity during AO only. Therefore, it is unclear how only AO trial effects on the result of this study. I think you need discuss about effect of AO on the result of this study in disussion, or measure the ANS activity during AO in isolation.

In this study, the ANS activity was measured after each MI session for 30 seconds. However, immediately after MI session, the LF/HF ratio as index of sympathetic nerve activity was returned at rest level (Bunno et al., 2015). Additionally, many authors measured the ANS activity concurrent with MI session (Decety et al., 1991, 1993; Demougeot et al., 2009; Williamson et al., 2002). Please explain why the author measured the ANS activity after MI session.

Validity of the findings

no comment

Additional comments

This study was well designed and described. However, as above, there are several points should be described in detail. Also, I suggest you discuss what clinical significance does the result of this study has.

---

## Round 0.2 · accepted · Accept

Dear Authors It is my pleasure to inform that your manuscript is accepted to be published in Peer J subjected to further administrative process.Thank you and Congratulations!

# ·

Basic reporting

no comment

Experimental design

no comment

Validity of the findings

no comment

Additional comments

I think that revision has been made sufficiently for comments. In future, I would like you to promote the research to resolve the limitation.